# Serotonergic amplification of odor-evoked neural responses maps onto flexible behavioral outcomes

Yelyzaveta Bessonova, Baranidharan Raman*

Department of Biomedical Engineering, Washington University in St. Louis, St. Louis, United States

**Abstract** Behavioral responses to many odorants are not fixed but are flexible, varying based on organismal needs. How such variations arise and the role of various neuromodulators in achieving flexible neural-to-behavioral mapping is not fully understood. In this study, we examined how serotonin modulates the neural and behavioral responses to odorants in locusts (*Schistocerca americana*). Our results indicated that serotonin can increase or decrease appetitive behavior in an odor-specific manner. On the other hand, in the antennal lobe, serotonergic modulation enhanced odor-evoked response strength but left the temporal features or the combinatorial response profiles unperturbed. This result suggests that serotonin allows for sensitive and robust recognition of odorants. Nevertheless, the uniform neural response amplification appeared to be at odds with the observed stimulus-specific behavioral modulation. We show that a simple linear model with neural ensembles segregated based on behavioral relevance is sufficient to explain the serotonin-mediated flexible mapping between neural and behavioral responses.

**\*For correspondence:**
barani@wustl.edu

**Competing interest:** The authors declare that no competing interests exist.

## eLife assessment

This **useful** work shows that the experimental application of serotonin to locust antennal lobes induces an increased feeding-related response to some odorants (even in food-satiated animals). To explain how the odorant-specific effects are seen despite similar consequences of 5-HT modulation on all projection neuronal types, the authors propose a simple quantitative model built around projection with different downstream connections. While they are consistent with the authors' conclusions, the current panel of experiments is **incomplete** and additional future work will be required to fully support the conclusions the authors currently draw from their observations.

## Introduction

Often the same sensory stimulus can trigger different behavioral responses within a single organism. For example, appetitive odorants that are attractive while hungry may not drive the same behavioral response after feeding to satiety (*Vogt et al., 2021*). How neural circuits process sensory stimuli to flexibly drive varying outcomes is not fully understood. It is hypothesized that different neuromodulators should be involved in mediating state-dependent changes in neural and behavioral responses (*Lizbinski and Dacks, 2017*; *Sayin et al., 2018*). However, whether neuromodulation globally changes responses to sensory cues (i.e., nonspecific increases or decreases) or mediates selective alterations in sensory input-driven behavior, and how this is achieved, is not clear (*Anton and Rössler, 2021*).

In insects, serotonin (5HT) is a key neuromodulator that is linked with the regulation of many behaviors, including feeding (*Tierney, 2020*), socialization (*Ries et al., 2017*; *Rogers et al., 2004*; *Anstey et al., 2009*), aggression (*Rillich and Stevenson, 2018*; *Dierick and Greenspan, 2007*), and mating

(*Chen et al., 2022*; *Pooryasin and Fiala, 2015*). Food intake is known to be highly dependent on the serotonin levels in the brain of an organism. Elevated levels of serotonin have been shown to decrease the time spent feeding and the amount of consumed food in blowflies and flesh flies (*Dacks et al., 2003*; *Long and Murdock, 1983*). In *Drosophila* larvae and adults, inhibition of neural serotonin synthesis or global increase of serotonin levels increases or suppresses food intake, respectively (*Neckameyer, 2010*; *Eriksson et al., 2017*; *Majeed et al., 2016*). In locusts, serotonin is linked to triggering phenotypical plasticity. A spike in serotonin levels in the locust brain is highly correlated with the solitary stage (*Guo et al., 2013*), whereas high levels of serotonin in the locust thoracic ganglion are known to trigger the transformation of solitary animals into gregarious ones (*Rogers et al., 2004*; *Anstey et al., 2009*). Serotonin modulation has also been linked with increased activity and aggression in crickets and flies (*Dierick and Greenspan, 2007*, *de Boer et al., 2016*), as well as a reduction in courtship behaviors, like male wing extension toward females and mating in flies (*Pooryasin and Fiala, 2015*).

How serotonin alters sensory processing to drive these behavioral changes is yet to be completely resolved. In olfaction, serotonin is known to modify the processing of odor-driven neural signals right from the periphery. In both vertebrates and invertebrates, the serotonergic release is known to reduce olfactory sensory neuron output to the following circuit through presynaptic GABAergic inhibition (*Gaudry, 2018*; *Petzold et al., 2009*; *Lv et al., 2023*). In contrast, exogenous serotonin is known to increase the odor-evoked responses of second-order neurons in the insect antennal lobe and vertebrate olfactory bulb (*Gaudry, 2018*; *Brill et al., 2016*). The serotonin-mediated increase in odor-evoked responses has been reported to be odor-specific and hypothesized to enhance the sensitivity to odorants in a state-dependent manner (*Dacks et al., 2009*).

How are odor-evoked neural responses modulated to produce flexible, odor-specific changes in behavioral outcomes? In this study, we examined this question in the locust olfactory system. We show that serotonin can increase or decrease innate appetitive behavioral responses in an odor-specific manner. In contrast, exogenous serotonin increased the strength of odor-evoked neural responses for all odorants without altering the temporal processing features or the ensemble response fidelity. We present a simple model to map the serotonin-mediated amplification of neural responses onto odor-specific changes in behavioral outcomes. Finally, we examine the relevance of these findings for modulating hunger-state-dependent modulation of appetitive responses in locusts. In sum, our results provide a more systems-level view of how a specific neuromodulator (serotonin) alters neural circuits to produce flexible behavioral outcomes.

## Results

### Serotonin modulates appetitive behavior in an odor-specific manner

We began by examining how serotonin modulates odor-driven innate behavioral responses in locusts (*Schistocerca americana*). In this assay, starved locusts opened their sensory appendages close to the mouth, called maxillary palps, when encountering certain food-related odorants (*Simões et al., 2011*; *Saha et al., 2013b*; *Nizampatnam et al., 2018*; *Chandak and Raman, 2021*). We examined the palp-opening responses (POR) for an odor panel including four odorants: hexanol (HEX), benzaldehyde (BZA), linalool (LOOL), and ammonium (AMN). Note that hexanol (green) is a green-leaf volatile (*Bertrand et al., 2021*), and benzaldehyde (blue) is a putative locust aggregation pheromone *Torto et al., 1994*. Whereas linalool (red) is used in many pesticides (*Beier et al., 2014*). Hence, this panel included odorants that have diverse behavioral preferences (*Chandak and Raman, 2021*). Each odorant was delivered at 1% v/v concentration and presented for 10 trials or repetitions in a pseudo-randomized order (see 'Methods').

We used a binary metric to categorize the presence or absence of PORs (*Figure 1A*). The response matrix across locusts (rows) and trials (columns) is summarized in *Figure 1B*. It is worth noting that HEX evoked supra-median (0.62) PORs, while BZA and LOOL have a response close to the median (0.4) POR level (mean POR = 0.43 across odorants and locusts). In contrast, AMN elicited sub-median (0.29) PORs across locusts. We also examined the PORs before and after serotonin (5HT) injection for each locust (see 'Methods'). Notably, we found that the probability of PORs (*Figure 1C*) changed after 5HT injection but only for a subset of odorants (HEX and BZA). Intriguingly, the PORs to LOOL decreased after 5HT injection. Injection of saline or merely repeating the same set of odorants after a

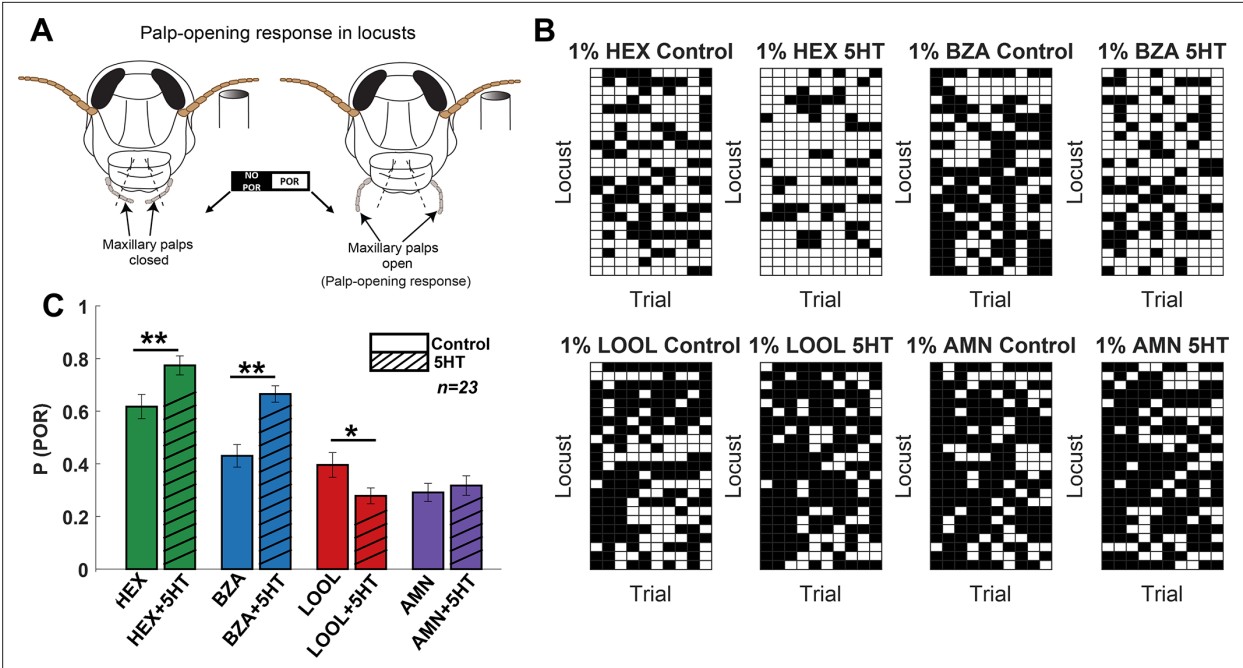

**Figure 1.** Serotonin modulates innate appetitive behavioral responses in an odor-specific manner. (**A**) A schematic of the locust palp-opening response (POR) is shown. Starved locusts (>24 hr) were presented with a panel of four odorants (hexanol [HEX]; benzaldehyde [BZA]; linalool [LOOL]; ammonium [AMN]) at 1% v/v dilution. Each locust was presented with 10 trials of each odorant. The odor pulse duration was 4 s, and the time between two consecutive odor exposures (inter-trial interval [ITI]) was 56 s. Movement of the palps during the odor presentation was identified as a positive POR. The presence or absence of a POR was noted for each trial. (**B**) A summary of trial-by-trial PORs for each locust is shown. Each trial was categorized by the presence (white box) or absence (black box) of a POR. Each row represents the PORs recorded from a single locust, and each column indicates a specific trial. PORs of 23 locusts were recorded and summarized as a response matrix. The POR matrix for the same set of locusts before and after 5HT injection is shown to allow comparison. (**C**) The PORs before and after (5HT) serotonin injection are summarized and shown as a bar plot for all four odorants in the panel. Striped bars signify the data collected after the 5HT injection. Significant differences are identified in the plot (one-tailed paired-sample *t*-test; *p<0.05; **p<0.01; standard paired-sample *t*-test).

The online version of this article includes the following figure supplement(s) for figure 1:

**Figure supplement 1.** Saline injection control experiments.

**Figure supplement 2.** Serotonin does not alter the palp-opening responses (PORs) evoked by paraffin oil (i.e., the solvent used to dilute odorants).

**Figure supplement 3.** Pictures showing the behavioral experiment setup and representative palp-opening responses in a locust.

**Figure supplement 4.** Palp-opening responses (PORs) patterns to different odorants remain consistent following serotonin introduction.

3-hr time window (similar time frame as before and after 5HT injection) did not produce any significant change in the PORs (*Figure 1—figure supplements 1 and 2* show PORs to paraffin oil before and after 5HT application). These results suggest that 5HT altered appetitive behavioral responses in an odor-specific manner.

## Serotonin enhances olfactory arousal to odorants

Next, we wondered whether serotonin introduction altered the behavioral dose–response relationship. To examine this, we recorded PORs to the same four odorants at widely different concentrations (*Figure 2*; spanning four log units of concentration). First, even without serotonin application, it can be noted that PORs to the odorants tended to increase as a function of odor intensity. Further, the increase was more significant for HEX and BZA, two odorants that generally elicited more PORs. However, note that there was a detectable decrease in PORs at the highest concentration of HEX and BZA.

Injection of serotonin increased the probability of generating POR for all concentrations of HEX and BZA. In contrast, changes in PORs elicited by LOOL and AMN were only modestly modified after 5HT introduction. Hence, these findings imply that serotonin enhanced appetitive behavioral responses over a wide range of concentrations but only for a subset of odorants (HEX and BZA).

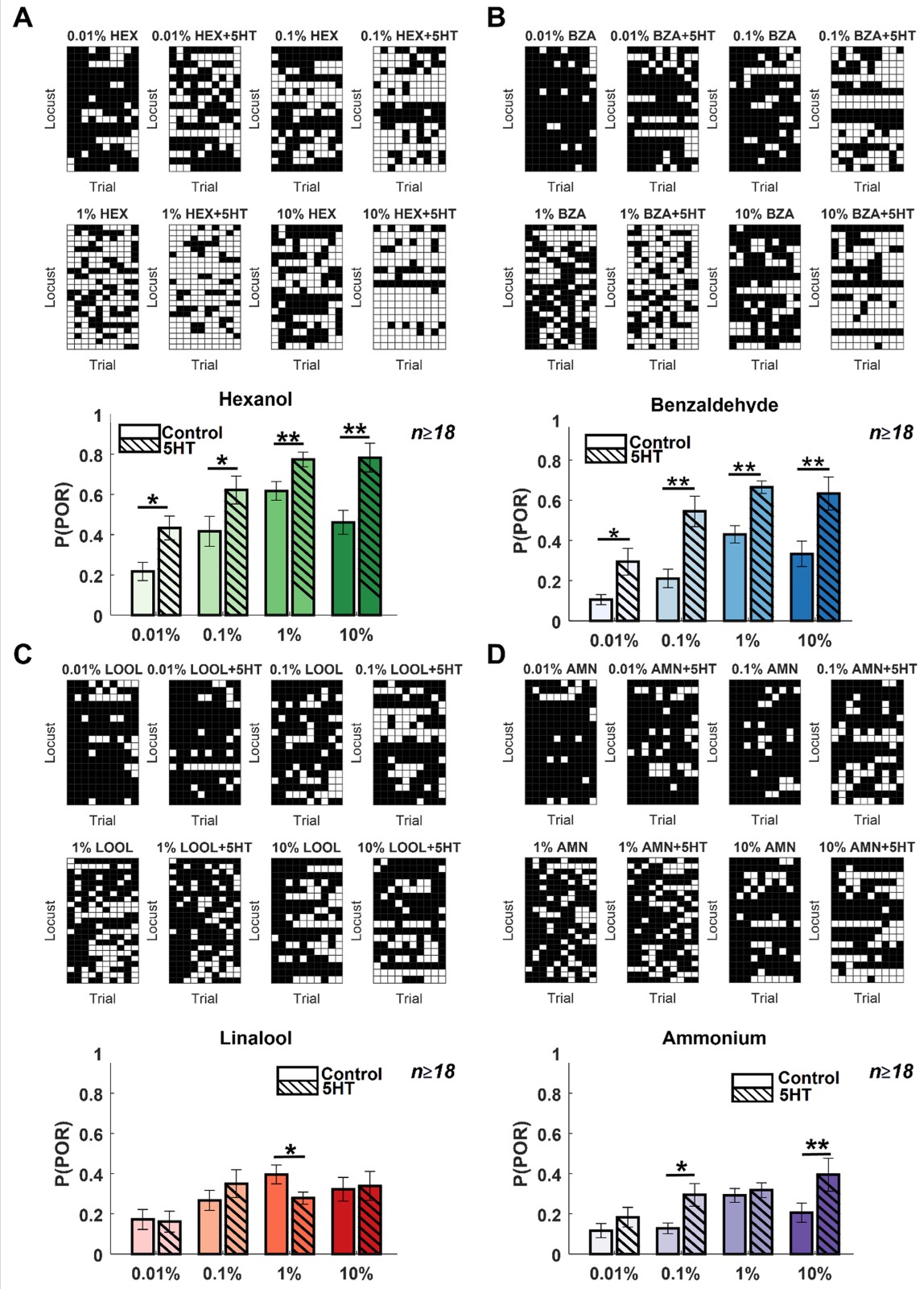

**Figure 2.** Serotonin alters the dose–response relationship for select odorants. (**A**). POR responses as a function of varying odor concentrations for hexanol are shown. Responses before and after 5HT introduction are shown for each odor concentration. Each trial was categorized by the presence (white box) or absence (black box) of a POR. Each row represents the PORs recorded from a single locust, and each column indicates a specific trial. (Bottom panel) The left solid bar shows the p(POR) for each odorant as a function of concentration before serotonin injection. The right striped bar

*Figure 2 continued on next page*

*Figure 2 continued*

summarizes p(POR) after serotonin injection for the same set of locusts (*p<0.05; ** p<0.01 standard one-tailed paired-sample t-test). (**B**) Similar plots as in panel A but showing POR responses to various concentrations of benzaldehyde before and after 5HT introduction. (**C**) Similar plots as in panel A but showing POR responses to various concentrations of linalool before and after 5HT introduction. (**D**) Similar plots as in panel A but showing POR responses to various concentrations of ammonium before and after 5HT introduction.

Furthermore, the reductions in behavioral responses at the highest intensity of HEX and BZA were no longer detectable or significant after 5HT treatment. In sum, these results indicate that the overall arousal to appetitive odorants is enhanced after serotonin application, and increases in behavioral response are maintained over a wide range of concentrations.

## Serotonin alters the spontaneous activity of projection neurons in the antennal lobe

Next, we investigated the neural basis of the observed serotonergic modulation of behavioral outcomes. To assess this, we intracellularly monitored the spiking activity of projection neurons (PNs)

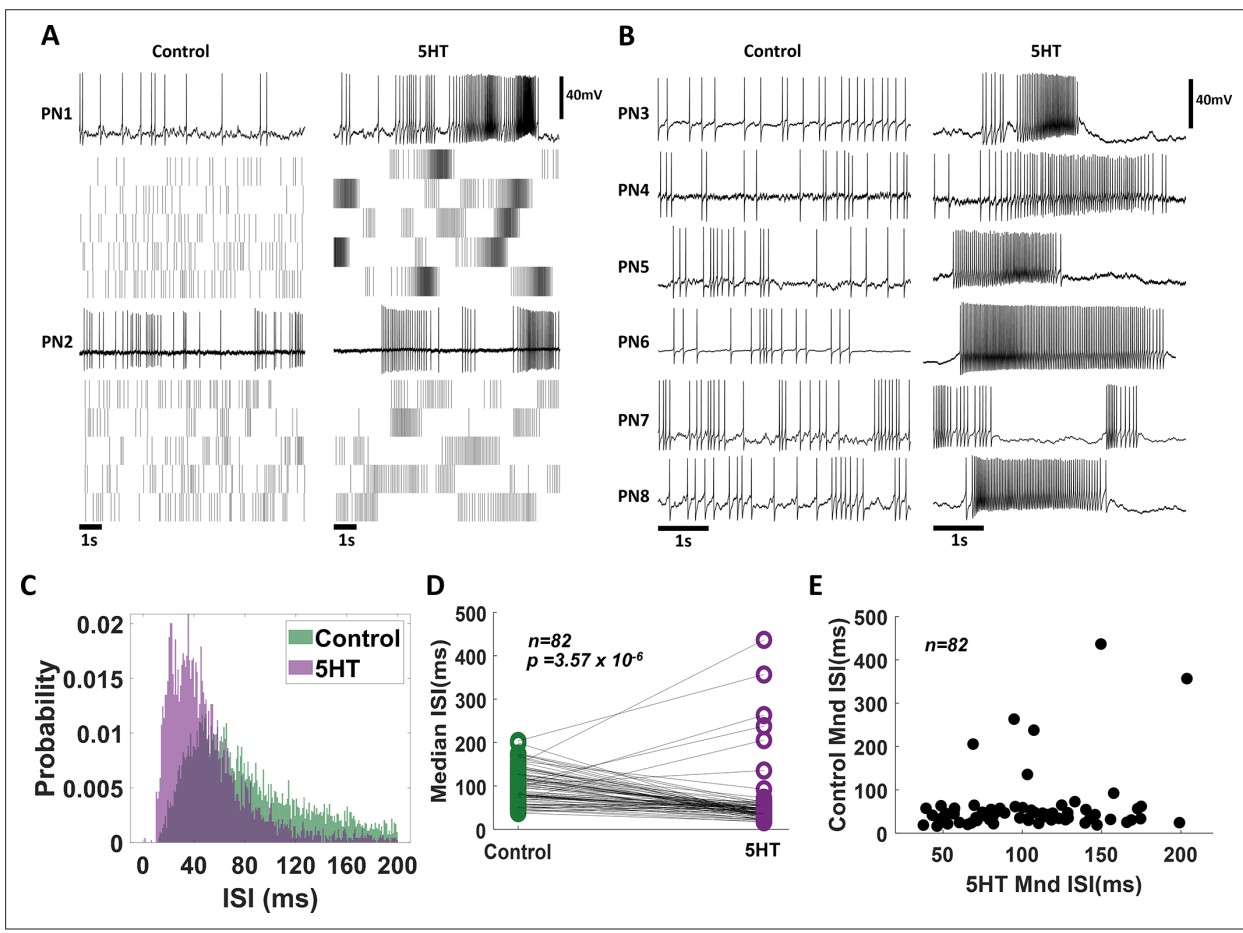

**Figure 3.** Exogenous serotonin induces bursting behavior in antennal lobe projection neurons (PNs). (**A**) Representative intracellular recordings showing membrane potential fluctuations as a function of time for two separate PNs in the locust antennal lobe. A 10 s window when no odorant was presented is shown. Raw voltage traces are shown during the first trial, and spiking rasters are shown for the subsequent trials. Firing patterns before (left) and after (right) serotonin application are shown for comparison. Note, that the spiking activity becomes more bursty after the 5HT application. (**B**) Firing patterns for a larger set of PNs are compared before (control) and after (5HT) serotonin. Changes in PN excitability are observed in all recorded PNs. (**C**) The distribution of Inter-spike intervals (ISIs) across PNs is shown before serotonin application (control; in green) and after serotonin application (purple). Note that the purple histogram is taller and shifted to the left, indicating shorter gaps between consecutive action potentials. (**D**) Comparison of the median ISIs for each individual PN before (control) and after 5HT application. The black line connects median ISI values for a single PN in control and 5HT conditions. Note the majority of the black lines are tilted downward, indicating a reduction in the gap between spikes. (**E**) Median ISI values before and after the 5HT application are plotted for each PN.

in the antennal lobe. We first compared the spontaneous spiking activities of the same neuron before and after (5HT) serotonin treatment (*Figure 3A* vs. *Figure 3B*). Note that the baseline spontaneous firing patterns in individual projection neurons changed after bath application of serotonin. The PNs fired spontaneous bursts of spikes after serotonin exposure (*Figure 3A* [before 5HT] vs. *Figure 3B* [after 5HT]). To quantify these differences, we computed the inter-spike interval (ISI) distributions across all PNs (*Figure 3C–E*; n = 82 PNs). The ISI distributions after 5HT application showed a left shift, indicating that pairs of consecutive spikes occurred in quick succession (*Figure 3C*). Consistent with these results, the median values for the ISI distribution reduced significantly for each PN recorded (*Figure 3D*). Finally, we plotted the median ISI before 5HT (control; along the x-axis) against median ISI values after 5HT application (y-axis) for each PN (*Figure 3E*). As can be noted, each point represents a single PN. Clustering closer to the X-axis indicates that serotonin altered the excitability of PNs and induced bursts of spikes.

In addition, we found that the total spiking activity in individual PNs monotonically increased with the magnitude of the current injection (*Figure 4—figure supplement 2*). However, after serotonin injection we found that the spiking activity remained relatively stable and did not systematically vary with the magnitude of the current injection. While the changes in odor-evoked responses may incorporate both excitability changes in individual PNs and recurrent feedback inhibition through GABAergic LNs, these results from our current injection experiments unambiguously indicate that there are changes in excitability at the level of individual PNs.

## Serotonin modulates odor-evoked response intensity but not timing

As noted, serotonin modulated the excitability of all individual PNs we recorded. How are the odor-evoked responses modulated, and does serotonergic modulation confound the information about odor identity? To examine this, we analyzed the odor-evoked responses of 82 PN-odor combinations. In addition to the four odorants examined in behavioral experiments, the odor panel used included three additional odorants (4-vinyl anisole, 1-nonanol, and octanoic acid). Consistent with previous observations, PNs responded to odorants by increasing their spiking activity either during odor presentation (ON response) or after stimulus termination (OFF response) (*Figure 4A*). These odor-evoked response patterns were consistent across trials.

We examined whether the odor-evoked response timing (ON vs. OFF responses) was preserved after serotonin application (*Figure 4A*). As can be noted, the stimulus-evoked ON and OFF responses in these four representative PNs remained intact after 5HT application. We found that for a majority of the PNs in our dataset that exhibited either an ON or OFF response, the response timing was maintained after 5HT application (*Figure 4A–C*). Notably, ON responses tended to be more robustly maintained after the 5HT application compared to the OFF responses (*Figure 4C and F*). Furthermore, most of the nonresponsive PNs continued to remain inactive during the odorant presentation after serotonin introduction.

What variations in spiking patterns were observed after the 5HT application? Our results indicate that the total number of spikes (i.e., response magnitude) evoked by an odorant increased after 5HT application during both ON and OFF time periods (*Figure 4D and F*). However, mere introduction of saline or the solvent in which serotonin was diluted (HCl) did not alter the PN responses (*Figure 4— figure supplement 1*). In sum, these results indicate that serotonin modulates the excitability of individual PNs, but only those PNs that were activated by an odorant tended to increase their response magnitude. Further, the response timing (ON vs. OFF periods) was robustly maintained across PNs.

## Robust encoding of odor identity

Is the identity of the odorant robustly encoded by the ensemble-level odor-evoked responses? To understand this, we first visualized the overall PN responses across all neuron-odor combinations in our dataset (*Figure 5A*). Consistent with the individual PN analyses, the changes in spontaneous activities and stability of spiking responses during the odor presentation period can be readily observed. To qualitatively analyze this, we visualized the neural activity using principal component analysis (PCA) dimensionality reduction. The 82-dimensional PN spike counts in 50 ms time bins were projected onto the top two eigenvectors of the response covariance matrix and connected in the order of occurrence to generate closed-loop trajectories (*Figure 5B*). Neural response trajectories during (ON) and after (OFF) odor presentation periods activated different subsets of PNs and therefore did not have much

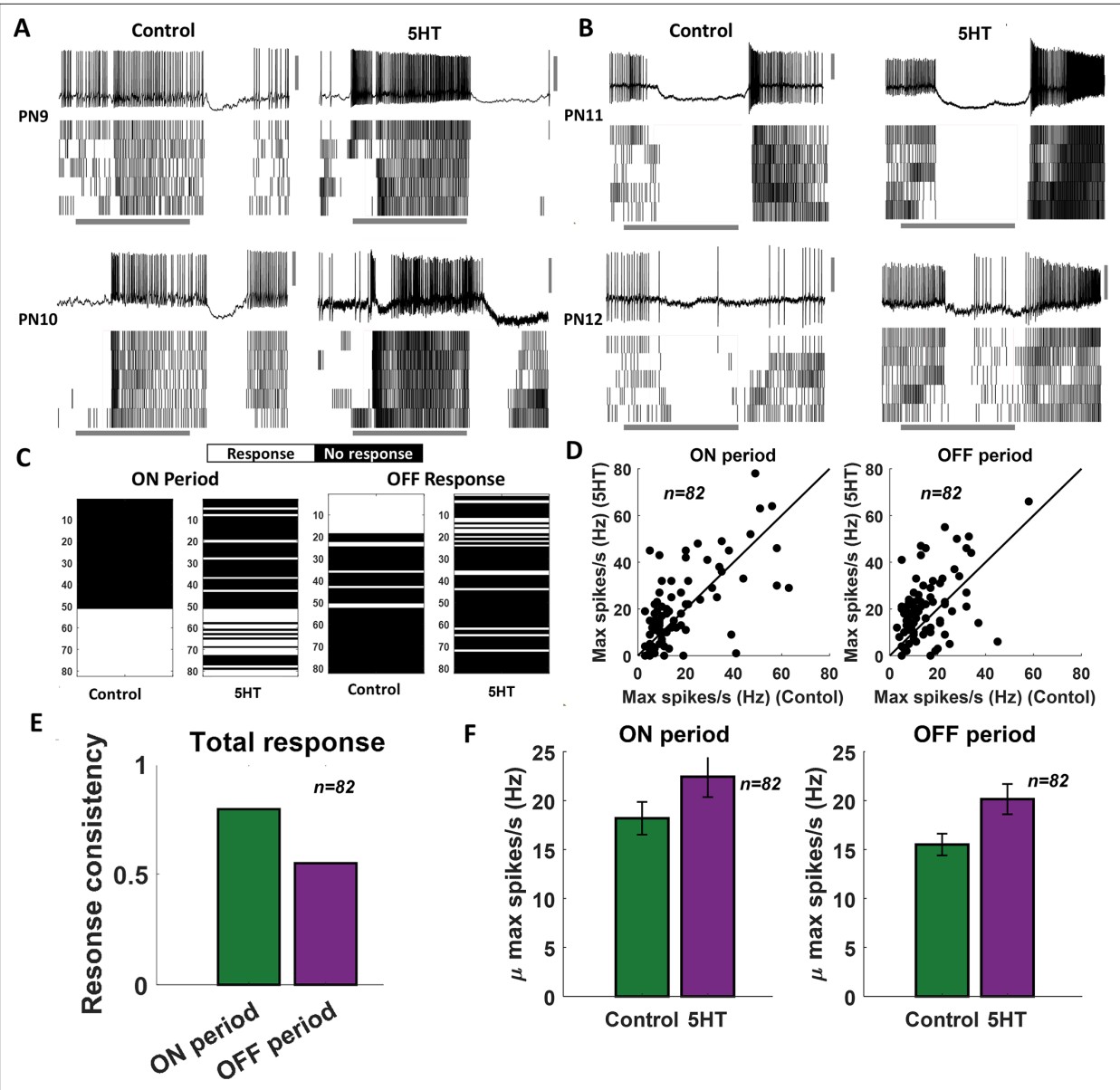

**Figure 4.** Serotonin alters the magnitude of odor-evoked spiking activity but not its timing. (**A, B**) Representative plots of the odor-evoked ON responses (PN9-10) and OFF responses (PN11-12) for five trials are shown. The first trial is shown as a raw voltage trace. Spiking activities during all five trials are shown as a raster plot. The horizontal gray bar indicates the 4 s odor delivery period. The vertical gray scale bar identifies 40 mV. (**C**) A binary plot categorizing projection neuron (PN) activity as responsive or nonresponsive during odor presentation (ON responses) or after odor termination (OFF responses). Response categorization before and after 5HT application is shown for each PN to examine response robustness. (**D**) Left panel: peak spiking activity for each PN during odor presentation in the control condition and after 5HT application. Right panel: comparing peak spiking activity observed during the OFF period. (**E**) Fraction of PNs that maintain their response or lack of response to an odorant before and after 5HT application are quantified during ON and OFF periods. (**F**) Left panel: mean odor-evoked spiking activity across PNs during odor presentation in the control condition and after 5HT application is compared. Error bars indicate SEM. Right panel: similar plot comparing mean spiking activity across cells during the OFF period.

The online version of this article includes the following figure supplement(s) for figure 4:

**Figure supplement 1.** Electrophysiology control experiments.

**Figure supplement 2.** Current-injection-induced spiking activity in individual projection neurons (PNs) is altered after serotonin application.

**Figure supplement 3.** Odor-evoked activity in the antennal lobe local neurons versus projection neurons.

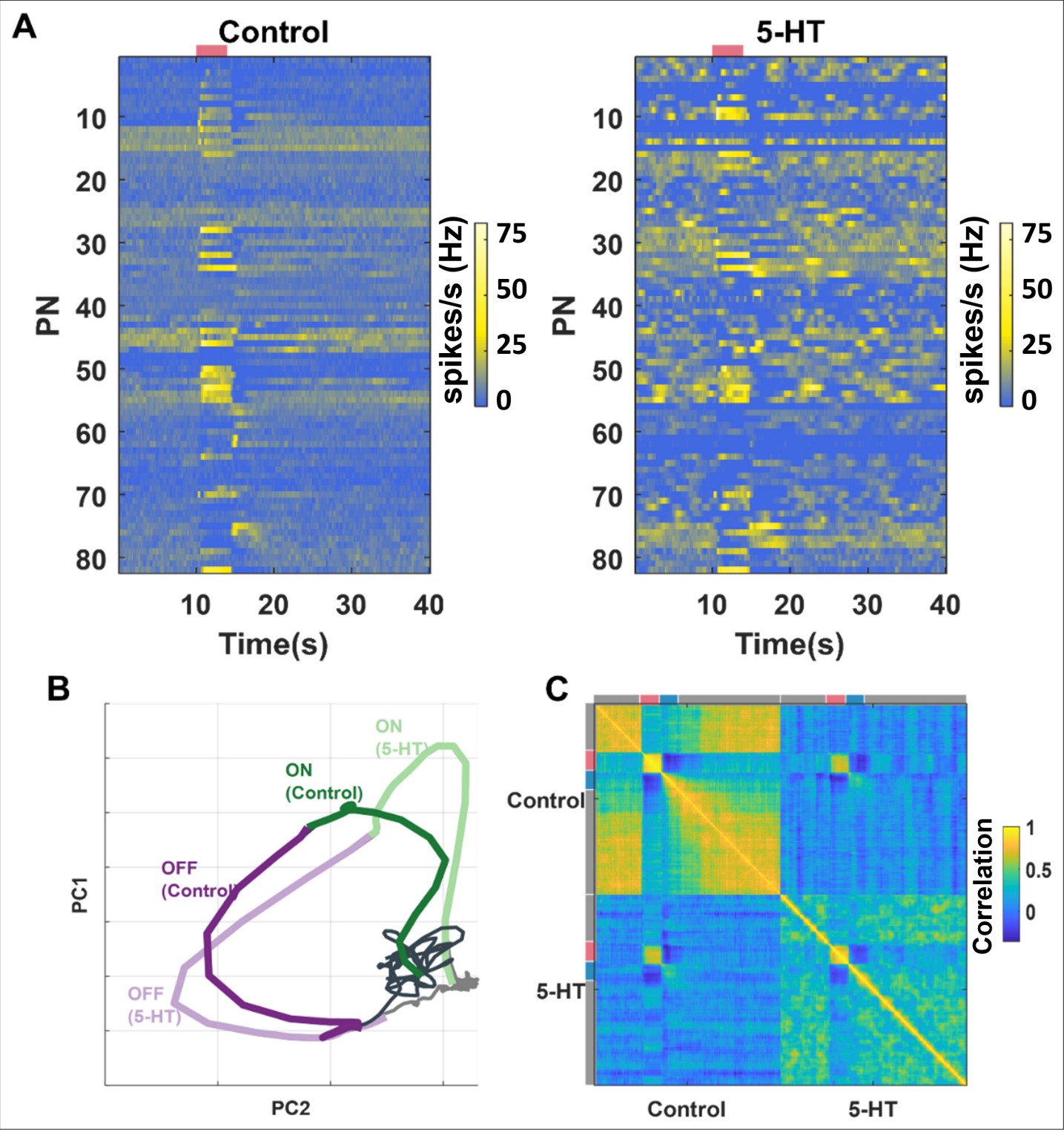

**Figure 5.** Ensemble-level odor-evoked response patterns robustly maintain odor identity after 5HT treatment. (**A**) Trial-averaged spiking activity as a function of time is shown for 82 neurons. The hotter color identifies the higher average firing rates per bin (200 ms). Each row represents one projection neuron (PN), each column represents one-time bin. The red bar identifies the odor presentation time period. The heatmap on the left shows the PN activity matrix before 5HT application, and the heatmap on the right shows the PN activity matrix (neurons sorted in the same order) after (5HT) exposure. (**B**) PN odor-evoked responses (n = 82 PN-odor combinations) are visualized after dimensionality reduction using principal component analysis (PCA). The neural responses were binned in 50 ms windows and projected onto the top two eigenvectors of the response covariance matrix and connected in the order of occurrence to generate the response trajectory shown. Neural response trajectories evoked during the OFF period are shown

*Figure 5 continued on next page*

*Figure 5 continued*

in purple, and ON response trajectories are shown in green. Darker colors indicate response trajectories before 5HT treatment, and lighter shades show neural trajectories after 5HT treatment. (**C**) A correlation matrix summarizing the similarity between each 82-dimensional PN activity vector with all other response vectors is shown. Different time segments (spontaneous [gray], odor ON [red], and odor OFF [blue]) are indicated along the x- and y-axes. The hotter color indicates a higher correlation.

overlap during these time periods. We also plotted the neural response trajectories before and after 5HT application to allow comparison. Our results indicate that the ON response trajectories before and after 5HT application overlapped with each other, indicating that the ensemble activity across the 82 PNs was similar before and after serotonin application. Similar results were also observed during the OFF-response window.

To further support the qualitative dimensionality reduction analysis, we performed a quantitative correlation analysis using high-dimensional PN activity vectors (*Figure 5C*). As can be noted, the structure of spontaneous activity before and after serotonin changed, and as a result, the correlation between them became negative. Consistent with the results from the PCA, we found that the odor-evoked responses before and after 5HT application were highly correlated.

Taken together, these results indicate that the identity of the odorant is robustly maintained by ensemble neural activity. Both ON and OFF responses continue to preserve the information they carry about the odorant.

## A simple linear model explains serotonergic modulation of neural-behavioral mapping

Our results indicate that serotonin modulated behavioral responses in an odor-specific manner. However, neural responses elicited by all odorants increased (*Figure 6A*). These results seem to be at odds with each other. To gain mechanistic insights regarding how 5HT uniformly amplifies neural responses while also generating odor-specific changes in behavioral outcomes, we performed a simple linear regression (*Figure 6B*). We used a previously published extracellular recording dataset (*Chandak and Raman, 2021*) consisting of odor-evoked responses of 89 PNs to HEX, BZA, and LOOL to build the model. The output of the model was the amount of increase or decrease in PORs compared to the overall mean response levels across all odorants used in our behavioral experiments (see 'Methods'). Note that this model is equivalent to assigning a weight to each projection neuron and using the weighted sum of projection neuron responses to generate the observed POR output.

We hypothesized that the antennal lobe projection neurons can be divided into two nonoverlapping ensembles: Encoding Ensemble 1 is assigned mostly positive weights and Encoding Ensemble 2 are assigned negative weights (*Figure 6C*). Odorants that evoke supra-median PORs should activate more neurons in Encoding Ensemble 1, and those that produce sub-median-level PORs are expected to activate neurons in Encoding Ensemble 2 more. It is worth noting that this architecture is similar to having 'neuron–anti-neuron' pairs where one decoding neuron weighs the positive contribution to generate PORs, and the second decoding neuron collects the negative contributions and suppresses the same behavioral output. Such 'neuron–anti-neuron' pairs have been utilized for predicting overall motor outputs (*Wu et al., 2022*; *Saha et al., 2017*; *Britten et al., 1992*) and are highly consistent with the emerging view from other insect models that have shown mushroom body output neurons form segregated channels to drive opposing behaviors (*Aso et al., 2014a*; *Aso et al., 2014b*).

Our results indicate that this simple linear regression-based model was sufficient to robustly predict the observed PORs (*Figure 6D*). Furthermore, as expected, HEX and LOOL activated highly distinct neural ensembles. HEX-activated projection neurons received mostly positive weights and LOOL-activated neurons received negative weights (*Figure 6E*). Any increase in positively weighted PN responses (HEX response after 5HT) should increase the overall POR, whereas the increase in negatively weighted PN responses (LOOL after 5HT) should similarly decrease the behavioral output. Segregating odor encoding into behavior-specific channels in the antennal lobe would allow serotonin to amplify neural responses to all odorants, while still generating odor-specific increases or decreases in behavior.

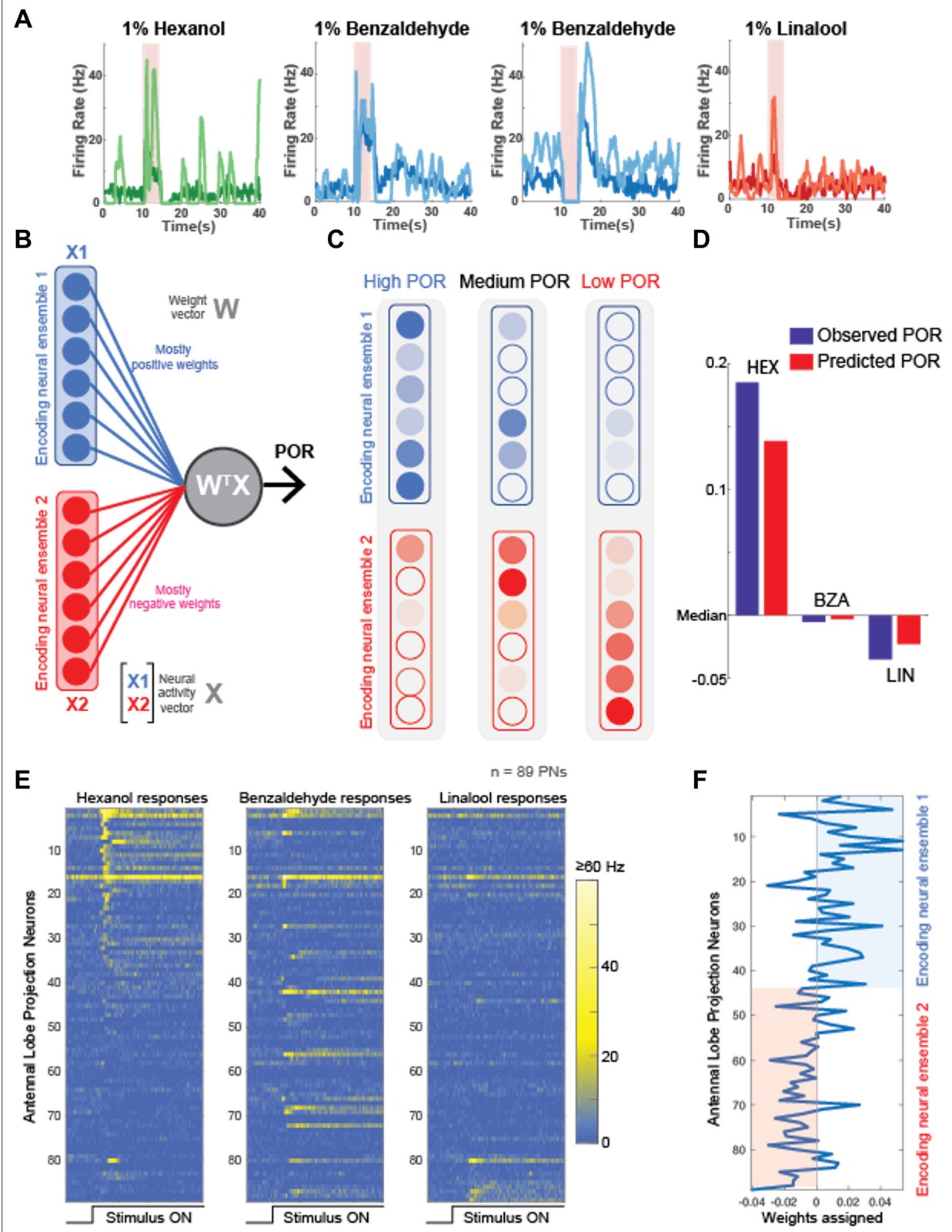

**Figure 6.** Neural data maps onto the behavioral results. (**A**) Peristimulus time histograms (PSTHs) of four representative projection neurons (PNs) are shown before (darker shade color) and after 5HT (lighter shade color) application. Note that 5HT increased overall response amplitudes to all odorants in the panel. (**B**) A schematic of the linear model used for predicting palp-opening responses (PORs) given neural data. Each neuron is assigned a weight. The weighted sum of PN activity is fit to POR values for hexanol (HEX), benzaldehyde (BZA), and linalool (LOOL) (see 'Methods'). The neurons

*Figure 6 continued on next page*

Figure 6 continued

were split into two ensembles based on their assigned weights. (**C**) Odorants that evoke stronger PORs are expected to activate more PNs that receive positive weights. In contrast, odorants that reduce POR output compared to the mean response are expected to activate more PNs that received negative weights. (**D**) Comparison of observed versus predicted POR values across locusts for the three odorants used in the model. (**E**) Odor-evoked responses of 89 PNs to HEX, BZA, and LOOL are shown. The PNs are ordered based on the difference in peak responses to HEX and LOOL (i.e., HEX-activated PNs are at the top and LOOL-activated neurons are at the bottom). (**F**) Weights assigned to each PN are shown. PNs are ordered the same as in panel (**E**).

The online version of this article includes the following figure supplement(s) for figure 6:

**Figure supplement 1.** Testing the generality of the proposed computational model.

## Hunger-state vs. serotonergic modulation of appetitive behavioral responses

Serotonin is regarded as one of the neuromodulators associated with feeding behaviors (*Tierney, 2020*). Therefore, we wondered whether serotonin modulates the behavioral appetitive responses of locusts in a hunger-state-dependent manner. To understand this, we first examined the PORs in locusts that were fed grass blades before the experiments (satiated) (*Figure 7*, *Figure 7—figure supplement 1*) Compared to the hungry locusts (*Figure 1*), the fed locusts responded less to HEX and BZA. However, consistent with results from the hungry locusts, the introduction of serotonin increased the appetitive PORs to HEX and BZA. Intriguingly, the appetitive responses of fed locusts treated with 5HT were comparable or slightly higher than the responses of hungry locusts to the same set of odorants. It is worth noting that responses to LOOL and AMN, non-food-related odorants with weaker PORs, reduced or remained unchanged in the fed locusts treated with 5HT. Therefore, we conclude that, like in many species, serotonin influences food-driven behaviors in locusts. However, since 5HT increased behavioral responses in both fed and hungry locusts, the precise role of 5HT modulation and whether it underlies hunger-state-dependent modulation of appetitive behavior still remains to be determined.

In sum, our results reveal a clear mapping between serotonergic modulation of odor-driven neural responses and how it regulates innate appetitive behavioral outcomes.

## Discussion

We examined how serotonin modulates odor-evoked neural and behavioral responses in locusts to a small but diverse panel of odorants. Our behavioral results revealed that serotonin increased innate appetitive responses to a subset of odorants (HEX and BZA). In contrast, responses to aversive or neutral odorants like LOOL (*Beier et al., 2014*) or AMN (*Delventhal et al., 2017*) decreased or had no significant change in their response levels. Consistent with the behavioral result, we found that the strength of the stimulus-driven responses increased in several PNs in the antennal lobe. However, the overall combination of neurons activated, and their temporal patterns of activation (ON vs. OFF responses), remained consistent. As a direct consequence, the identity of the odorant could be robustly maintained after exogenous serotonin introduction. Finally, fed locusts injected with serotonin generated similar appetitive PORs to odorants as observed in starved locusts.

Prior results from a number of invertebrate and vertebrate models have shown similar changes in odor-evoked neural responses in the antennal lobe (*Gaudry, 2018*; *Dacks et al., 2009*; *Kloppenburg and Hildebrand, 1995*; *Kloppenburg et al., 1999*; *Dacks et al., 2008*). The increase in the spiking activity of second-order neurons seems inconsistent with the serotonergic gating of sensory input through presynaptic inhibition (*Petzold et al., 2009*; *Lv et al., 2023*). Our results indicate that serotonin also modified the overall excitability of individual PNs and made them fire action potentials in bursts. Hence, it is possible that the increased neural sensitivity could compensate for the decreased input from sensory neurons. The change in input–output mapping as a result of serotonin introduction resulted in an increase in the behavioral response to different concentrations of HEX and BZA.

In contrast to the changes in the response strength, our results indicate that the timing of odor-evoked spiking activity was robustly maintained. As a result, the combination of neurons activated during and immediately after the presentation of the odorant remained similar before and after the introduction of serotonin. Hence, serotonergic modulation altered sensitivity to some odorants without altering the identity of the odorant. These results are consistent with prior imaging studies

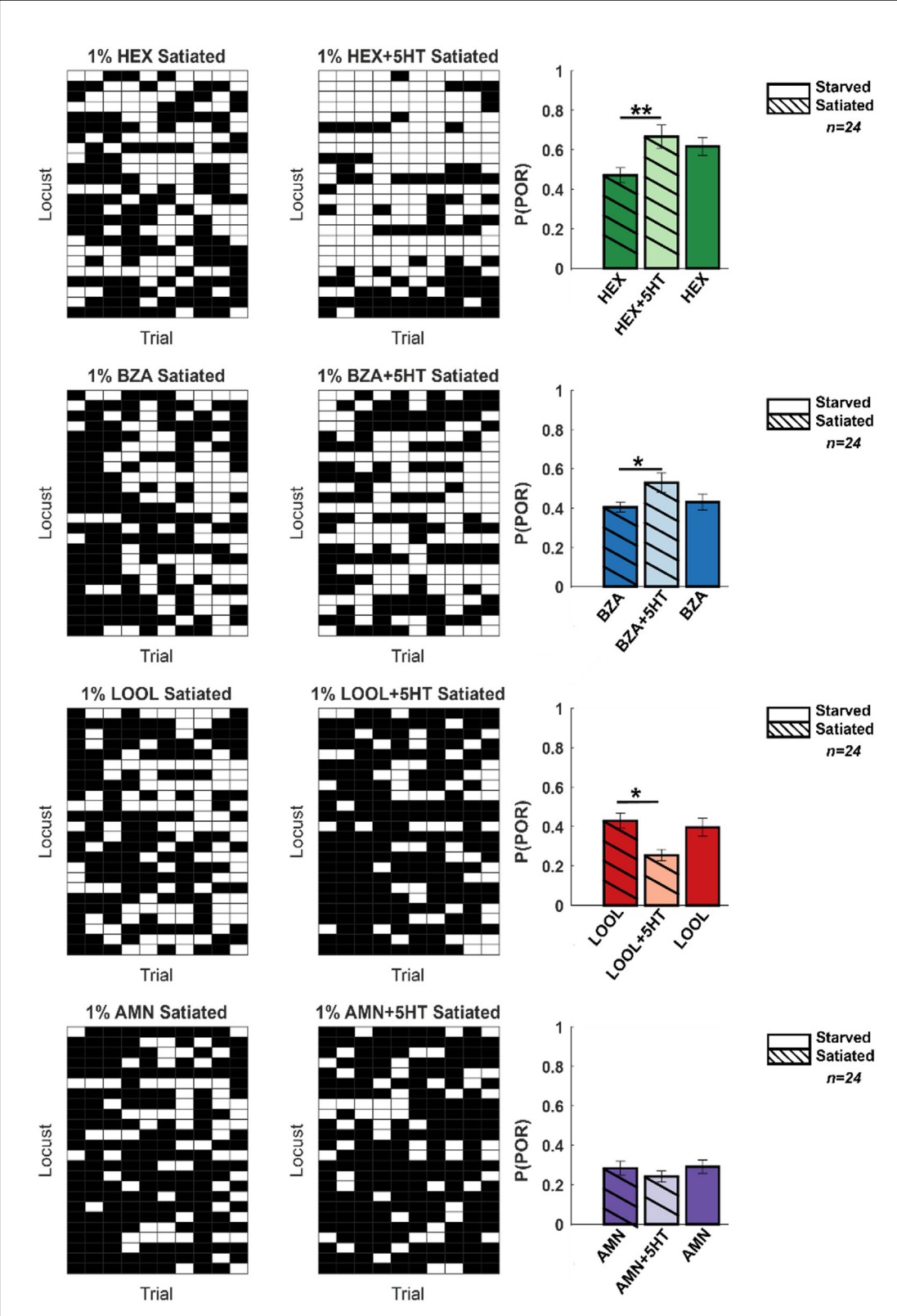

**Figure 7.** Hunger-state-dependent serotonergic modulation of appetitive behavioral responses. Left: a summary of trial-by-trial palp-opening responses (PORs) to the same four odorants used in **Figure 1**. The same convention was used in **Figure 1** (POR – white; no POR – black). Each row represents PORs recorded across a single locust in 10 trials. Twenty-four locusts were used. Each column represents a trial. Each odorant was presented at 1% v/v concentration. The POR matrix for the same set of locusts before and after 5HT injection is shown to allow comparison. Right: PORs are summarized

*Figure 7 continued on next page*

*Figure 7 continued*

and shown as a bar plot for all four odors for satiated locust (highlighted with lines), before (dark shade), and after 5HT injection (lighter shade). To allow comparison with POR in starved locust, results from *Figure 1* are re-plotted in solid bars without stripes Significant differences are identified in the plot (one-tailed paired-sample *t*-test; *p<0.05; **p<0.01; standard paired-sample *t*-test).

The online version of this article includes the following figure supplement(s) for figure 7:

**Figure supplement 1.** Summarized palp-opening responses (PORs) and shown as a bar plot for all four odors for satiated locust (highlighted with lines), before (dark shade), and after 5HT injection (lighter shade).

in flies that reported odor-specific changes in glomerular activity (*Dacks et al., 2009*). Further, this interpretation of our physiological results is consistent with the behavioral observation that serotonin altered response levels in an odor-specific manner.

However, behavioral responses could both increase or decrease depending on the odor identity. In contrast, our neural data indicated an enhancement for all odorants examined. This apparent mismatch between neural and behavioral responses could be resolved using a very simple linear regression model. In the model, a subset of neurons that were activated strongly by odorants with stronger PORs received positive weights. In contrast, the subset of neurons that responded to odorants that evoked fewer PORs received negative weights. Notably, the segregation of neural subsets based on the behavioral relevance and the opposing weights assigned to them was sufficient to produce odor-specific increase or decrease in behavioral PORs as observed during serotonergic modulation.

The model used for mapping neural responses onto behavior only required that odorants that evoke or suppress PORs activate distinct sets of neurons. It is worth noting that such segregation of neural responses could happen at any neural circuit along the olfactory pathway. Our extracellular recording data (*Figure 6E*) indicate that hexanol (high PORs) and linalool (low PORs) do indeed activate highly nonoverlapping sets of PNs in the antennal lobe. Hence, our results suggest that the segregation of neural activity based on behavioral relevance begins very early along the olfactory pathway.

While serotonin is implicated in a range of behaviors, the modulation of feeding behavior is widely conserved across invertebrates and vertebrates. Hence, we finally explored whether this role is conserved in locusts as well. We found that the appetitive response evoked by a food-related odorant (HEX; green-leaf volatile; *Bertrand et al., 2021*) was reduced in animals that were fed. Notably, we found that serotonin application in locusts that were fed recovered back their appetitive response to HEX. Earlier studies have shown that serotonergic neurons could mediate a hunger-state-dependent switch in behavioral response preference (attraction vs. repulsion) to the same food-related odorant in fruit fly larvae (*Vogt et al., 2021*). Our studies complement these findings and show how serotonin could modulate behavioral responses to different odorants in a stimulus-specific manner.

## Methods
### Animals
We used adult *S. americana* of both sexes from a crowded colony for our electrophysiology and behavioral experiments. Sixth-instar locusts were identified by the developed wings and soft cuticle in the neck area.

### Odor stimulation
The odor stimulus was delivered using a standard procedure (*Saha et al., 2013b*; *Chandak and Raman, 2021*; *Saha et al., 2015*; *Saha et al., 2013a*). Briefly, all odorants were diluted to their 0.01–10% concentration by volume (v/v) in mineral oil. Ammonium alone was diluted in distilled water. 20 ml of diluted odor solution was kept in 60 ml sealed bottles. During stimulus presentations, a constant volume (0.1 l/min) from the odor bottle headspace was displaced and injected into the carrier stream using a pneumatic pico pump (WPI Inc, PV-820). A vacuum funnel placed right behind the locust antenna ensured the removal of odor vapors.

## Odor panel

The odorants were selected based on chemical structure and ecological relevance. Therefore, we chose a very diverse odor panel that consisted of a food odorant (hexanol) (*Bertrand et al., 2021*), a sexual maturation pheromone (4-vinyl anisole) (*Assad et al., 1997*), and a putative aggregation pheromone (benzaldehyde) (*Torto et al., 1994*). In addition, we included an acid (octanoic acid), a base (ammonium), and an alcohol (1-nonanol) in our experiments. These odors were specifically selected based on our previous neural and behavioral data (*Saha et al., 2013b*; *Nizampatnam et al., 2018*; *Chandak and Raman, 2021*; *Nizampatnam et al., 2022*). For some experiments, we chose to focus on four odorants that had a diverse range of POR rates: supra-median (hexanol), median (benzaldehyde), and sub-median (linalool) POR levels.

## Behavioral experiments

Sixth-instar locusts of either sex were starved for 24–48 hr before the experiment or taken straight from the colony and fed blades of grass for the satiated condition. Locusts were immobilized by placing them in the plastic tube and securing their body with black electric tape (*Figure 1—figure supplement 3*). The head of the locusts, along with the antenna and maxillary palps, protruded out of this immobilization tube so that they could be freely moved. Note that the maxillary palps are sensory organs close to the mouth parts that are used to grab food and help with the feeding process. Locusts were given 20–30 min to acclimatize after placement in the immobilization tube.

Each locust was presented with one concentration of four odorants (hexanol, benzaldehyde, linalool, and ammonium) in a pseudorandomized order. The odor pulse was 4 s in duration and the inter-pulse interval was set to 60 s. The experiments were recorded using a web camera (Microsoft). The camera was fully automated with the custom MATLAB script to start recording 2 s before the odor pulse and end recording at odor termination. An LED was used to track the stimulus onset/offset. The PORs were scored offline. Responses to each odorant were scored a 0 or 1 depending on whether the palps remain closed or opened (*Figure 1A*). A positive POR was defined as a movement of the maxillary palps during the odor presentation time window as shown on the locust schematic (*Figure 1A*).

## Serotonin treatment

After the initial set of POR experiments, a 0.1 M serotonin solution was injected directly into the locust's head. The needle of a U-100 syringe was inserted slightly under the cuticle of the locust head, ~1 mm above the median ocellus. 1 ul of the solution was injected, and the opening was sealed with a small amount of melted dental wax. The locust was left to stabilize for 3 hr after the injection and before the second set of behavioral experiments.

We did electrophysiology experiments 5–10 min after bath application of 5HT. A longer delay (3 hr) was required for our behavioral experiments as the locusts tended to be a bit more agitated with larger spontaneous movements of palps as well as exhibited unprompted vomiting. However, we note that the POR patterns to various odorants evaluated 15 min and 3 hr after 5HT injection were consistent (*Figure 1—figure supplement 4*).

## Electrophysiological experiments

### Surgery

Sixth-instar locusts of either sex were used for these experiments. The legs and wings were removed and the locust was immobilized on a platform. The head was fixed with wax and a cup was built around the head to hold saline solution. The locust antennae were held in place using clear tubing and allowed to pass through the wax cup. The cuticle between the antenna was removed and the air sacs/trachea was removed to expose the brain. Additionally, the gut was removed and a metal wire platform was placed underneath the brain to lift and stabilize it. Finally, the transparent sheath was removed by carefully using sharp forceps. Locust's brains were super-fused with artificial saline buffer. A visual demonstration of this entire protocol is available online (*Saha et al., 2013a*).

### Electrophysiology

Intracellular recordings were performed using glass electrodes (resistance 8–15 MΩ) filled with intracellular saline (130 mM L-aspartic acid potassium salt, 2 mM $MgCl_2$, 1 mM $CaCl_2$, 10 mM HEPES, 10 mM EGTA, 2 mM $Na_2ATP$, 3 mM D-glucose, 0.1 mM cAMP; osmolarity ~315 mmol/kg; pH 7.0).

Glass electrodes were pulled using micropipette puller (Sutter Instrument, Novato, CA). Spontaneous firing, as well as real-time odor-evoked responses, was recorded in the current-clamp configuration. Each set of experiments consisted of five consecutive 40 s trials, with 20 s intervals in between for each odorant/odor concentration. Odor stimulation was performed at the 10th second of the trial for 4 s. Voltage signals were amplified (Axoclamp-2B, Molecular Devices) and saved using a custom MATLAB script.

After monitoring the responses to the odor panel, a serotonin solution was applied directly into the bath using a thin nozzle pipette. The serotonin solution (0.01 M serotonin hydrochloride in locust saline buffer) was made fresh before every experiment due to its light sensitivity. The same set of recordings and odor panel was repeated 5–10 min following serotonin application.

## Analysis

### Probability of POR calculation

PORs were scored in a binary fashion. PORs of locusts were summed across 10 trials and across all locusts. The probability for each odorant was calculated as follows:

$$p(POR) = \frac{Total\,Score_{odor}}{Total\,\#\,of\,locusts\,X\,10} \tag{1}$$

Significant differences between the PORs observed before and after serotonin application were calculated using a single-tailed paired-sample $t$-test ('$ttest$' built-in function in MATLAB).

### Electrophysiology data

In total, electrophysiological data from 82 odor-neuron combinations was obtained intracellularly and used for analyses. Each recording was preprocessed and converted into a response matrix. MATLAB built-in function 'findpeaks' was used for identifying action potentials. In total, our dataset includes recordings from 19 PNs. Seven PNs were tested on a panel of seven odorants (4-vinyl anisole, 1-nonanol, octanoic acid, HEX, BZA, LOOL, and AMN) and the remaining 12 were tested with the four main odorants used in the study (HEX, BZA, LOOL, and AMN). Note that in the locust antennal lobe only PNs fire full-blown sodium spikes. GABAergic local neurons only fire calcium spikelets and can be easily distinguished from PNs (*Figure 4—figure supplement 3*).

### PN response classification

We defined 4 s of odor presentation as an ON period, and the 4 s immediately following odor termination as an OFF period. For each PN, we used the mean baseline response during the pre-stimulus time period + 6.5 standard deviations of baseline activity as the threshold that needs to be exceeded to be classified as a response.

### Dimensionality reduction analysis

We used PCA to visualize ensemble PN activity. The spiking activity for each PN during 4 s of odor presentation was averaged across all five trials and binned in 200 ms nonoverlapping time bins. In this manner, we obtained a 164 PN × 200 time bin matrix for all odorants. The first 82 rows included PN responses before serotonin introduction and the other 82 rows included the responses of the same set of neurons after exogenous serotonin application. The response covariance matrix was calculated, and the data was projected onto the top two eigenvectors corresponding to the largest eigenvalues.

### Correlation analysis

Similar to the PCA, the data matrix used for this analysis was 164 PN × 200 time bins (40 s; 200 ms time bins). This included the 82-dimensional PN spike count vectors before, during, and after odor presentations. Spike counts were averaged across five trials.

The correlation analysis was done time-bin-by-time-bin. Each pixel or matrix element in time-bin-by-time-bin correlation plots (*Figure 5C*) indicates the correlation value between neural activity vectors

observed in the $i$th and $j$th time bins. All time-bin-by-time-bin correlation analyses were computed using high-dimensional response vectors. Correlations were calculated as

$$C_{ij} = \frac{cov(X_i, X_j)}{\sigma_i \cdot \sigma_j}$$ (2)

Here, $i$ and $j$ represent time bins, $X_i$, $X_j$ represents the population activity vector in the $i$th and $j$th time bin, respectively, $\sigma_i, \sigma_j$ are standard deviations of spiking activities during the $i$th and $j$th time bins, respectively.

## Linear regression model

We used a recently published dataset (*Chandak and Raman, 2021*) of extracellular PN responses to HEX, BZA, and LOOL to build our linear regression model. The input to the model was the spiking activity across the 89 neurons (89 PNs × 40 s; 89 × 800 response matrix for each odorant; each time bin was 50 ms in duration). The response matrix for the three odorants was concatenated (**X**; 89 × 2400). The PN weights were determined as follows:

$$W = (XX^T)^{-1}XY$$ (3)

X denotes the concatenated matrix of neural activity. Each column of **X** represents trial-averaged firing activity across 89 PNs in a 50 ms time bin. Neural responses before, during, and after the termination of all three odorants (HEX, BZA, and LOOL) were included. **Y** is a row vector with values set to mean subtracted p(POR) only during those time bins when HEX, BZA, and LOOL were presented (zeros otherwise). The generality of the model was tested using a larger dataset of neural and behavioral responses albeit without 5HT modulation (*Figure 6—figure supplement 1*).

## Code availability

The custom code used to generate figures in this article is publicly available along with the datasets in Figshare.

## Acknowledgements

We thank members of the Raman Lab (Washington University in St. Louis) and members of the Behavioral Plasticity Research Institute for their feedback on the manuscript and earlier presentation. We thank Pearl Olsen for insect care. We thank Ryan Sumida for writing the code to truncate and grab the behavioral video files. We thank Jacob Kelley for validating our Figshare data and code. This research was supported by NSF grant # 2021795 to BR.

## Additional information

### Funding

| Funder | Grant reference number | Author |
| --- | --- | --- |
| National Science Foundation | 2021795 | Baranidharan Raman |

The funders had no role in study design, data collection and interpretation, or the decision to submit the work for publication.

### Author contributions

Yelyzaveta Bessonova, Conceptualization, Formal analysis, Investigation, Methodology, Writing – original draft, YB and BR conceived the study and designed the experiments/analyses. YB performed all the behavioral and electrophysiology experiments and analyzed the data. BR developed the model. YB and BR wrote the paper. BR obtained the funds and supervised all aspects of the work; Baranidharan

Raman, Conceptualization, Formal analysis, Supervision, Funding acquisition, Investigation, Project administration, Writing – review and editing, YB and BR conceived the study and designed the experiments/analyses. YB performed all the behavioral and electrophysiology experiments and analyzed the data. BR developed the model. YB and BR wrote the paper. BR obtained the funds and supervised all aspects of the work

**Author ORCIDs**
Yelyzaveta Bessonova (iD) http://orcid.org/0009-0004-8313-3382
Baranidharan Raman (iD) https://orcid.org/0000-0002-7866-155X

Reviewer #1 (Public Review): https://doi.org/10.7554/eLife.91890.3.sa1
Reviewer #2 (Public Review): https://doi.org/10.7554/eLife.91890.3.sa2
Author response https://doi.org/10.7554/eLife.91890.3.sa3

## Additional files

### Supplementary files
• MDAR checklist

### Data availability
All data presented in this paper are publicly available in Figshare at https://doi.org/10.6084/m9.figshare.25365442. The custom code used to generate figures in this paper is publicly available along with the datasets in Figshare.

The following dataset was generated:

| Author(s) | Year | Dataset title | Dataset URL | Database and Identifier |
| --- | --- | --- | --- | --- |
| Bessonova Y, Raman B | 2023 | Serotonergic amplification of odor-evoked neural responses maps flexibly onto behavioral outcomes | https://doi.org/10.6084/m9.figshare.25365442 | figshare, 10.6084/m9.figshare.25365442 |

The following previously published dataset was used:

| Author(s) | Year | Dataset title | Dataset URL | Database and Identifier |
| --- | --- | --- | --- | --- |
| Chandak R, Raman B | 2023 | Neural manifolds for odor-driven innate and acquired appetitive preferences | https://doi.org/10.6084/m9.figshare.22656154.v1 | figshare, 10.6084/m9.figshare.22656154.v1 |

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
