## [Editor Report · eLife assessment]

This **useful** work shows that the experimental application of serotonin to locust antennal lobes induces an increased feeding-related response to some odorants (even in food-satiated animals). To explain how the odorant-specific effects are seen despite similar consequences of 5-HT modulation on all projection neuronal types, the authors propose a simple quantitative model built around projection with different downstream connections. While they are consistent with the authors' conclusions, the current panel of experiments is **incomplete** and additional future work will be required to fully support the conclusions the authors currently draw from their observations.

---

## [Referee Report · Reviewer #1 (Public Review)]

Summary:

This manuscript explores the impact of serotonin on olfactory coding in the antennal lobe of locusts and odor-evoked behavior. The authors use serotonin injections paired with an odor-evoked palp-opening response assay and bath application of serotonin with intracellular recordings of odor-evoked responses from projection neurons (PNs).

Strengths:

The authors make several interesting observations, including that serotonin enhances behavioral responses to appetitive odors in starved and fed animals, induces spontaneous bursting in PNs, directly impacts PN excitability, and uniformly enhances PN responses to odors.

Weakness:

The one remaining issue to be resolved is the theoretical discrepancy between the physiology and the behavior. The authors provide a computational model that could explain this discrepancy and provide the caveat that while the physiological data was collected from the antennal lobe, but there could be other olfactory processing stages involved. Indeed other processing stages could be the sites for the computational functions proposed by the model. There is an additional caveat which is that the physiological data were collected 5-10 minutes after serotonin application whereas the behavioral data were collected 3 hours after serotonin application. It is difficult to link physiological processes induced 5 minutes into serotonin application to behavioral consequences 3 hours subsequent to serotonin application. The discrepancy between physiology and behavior could easily reflect the timing of action of serotonin (i.e. differences between immediate and longer-term impact).

Overall, the study demonstrates the impact of serotonin on odor-evoked responses of PNs and odor guided behavior in locust. Serotonin appears to have non-linear effects including changing the firing patterns of PNs from monotonic to bursting and altering behavioral responses in an odor-specific manner, rather than uniformly across all stimuli presented.

---

## [Referee Report · Reviewer #2 (Public Review)]

Summary:

The authors investigate the influence of serotonin on feeding behavior and electrophysiological responses in the antennal lobe of locusts. They find that serotonin injection changes behavior in an odor-specific way. In physiology experiments, they can show that projection neurons in the antennal lobe generally increase their baseline firing and odor responses upon serotonin injection. Using a modeling approach the authors propose a framework on how a general increase in antennal lobe output can lead to odor-specific changes in behavior.

Strengths:

This study shows that serotonin affects feeding behavior and odor processing in the antennal lobe of locusts, as serotonin injection increases activity levels of projection neurons. This study provides another piece of evidence that serotonin is a general neuromodulator within the early olfactory processing system across insects and even phyla.

Weaknesses:

I still have several concerns regarding the generalizability of the model and interpretation of results. The authors cannot provide evidence that serotonin modulation of projection neurons impacts behavior.

The authors show that odor identity is maintained after 5-HT injection, however, the authors do not show if PN responses to different odors were differently affected after serotonin exposure.

Regarding the model, the authors show that the model works for odors with non-overlapping PN activation. However, only one appetitive, one neutral, and one aversive odor has been tested and modeled here. Can the fixed-weight model also hold for other appetitive and aversive odors that might share more overlap between active PNs? How could the model generate BZA attraction in 5-HT exposed animals (as seen in behavior data in Figure 1) if the same PNs just get activated more?

The authors should still not exclude the possibility that serotonin injections could affect behavior via modulation of other cell types than projection neurons. This should still be discussed, serotonin might rather shut down baseline activation of local inhibitory neurons - and thus lead to the interesting bursting phenotypes, which can also be seen in the baseline response, due to local PN-to-LN feedback.

The authors did not fully tone down their claims regarding causality between serotonin and starved state behavioral responses.

There is no proof that serotonin injection mimics starved behavioral responses.

---

## [Author Response]

[The following is the authors’ response to the current reviews.]

**Reviewer #1 (Public Review):**
Summary:This manuscript explores the impact of serotonin on olfactory coding in the antennal lobe of locusts and odor-evoked behavior. The authors use serotonin injections paired with an odorevoked palp-opening response assay and bath application of serotonin with intracellular recordings of odor-evoked responses from projection neurons (PNs).Strengths:The authors make several interesting observations, including that serotonin enhances behavioral responses to appetitive odors in starved and fed animals, induces spontaneous bursting in PNs, directly impacts PN excitability, and uniformly enhances PN responses to odors.Weaknesses:The one remaining issue to be resolved is the theoretical discrepancy between the physiology and the behavior. The authors provide a computational model that could explain this discrepancy and provide the caveat that while the physiological data was collected from the antennal lobe, but there could be other olfactory processing stages involved. Indeed other processing stages could be the sites for the computational functions proposed by the model. There is an additional caveat which is that the physiological data were collected 5-10 minutes after serotonin application whereas the behavioral data were collected 3 hours after serotonin application. It is difficult to link physiological processes induced 5 minutes into serotonin application to behavioral consequences 3 hours subsequent to serotonin application. The discrepancy between physiology and behavior could easily reflect the timing of action of serotonin (i.e. differences between immediate and longer-term impact).

For our behavioral experiments, we waited 3 hours after serotonin injection to allow serotonin to penetrate through the layers of air sacks and the sheath, and for the locusts to calm down and recover their baseline POR activity levels. For the physiology experiments, we noticed that the quality of the patch decreased over time after serotonin introduction. Hence, it was difficult to hold cells for that long. However, the point raised by the reviewer is well-taken. We have performed additional experiments to show that the changes in POR levels to different odorants are rapid and can be observed within 15 minutes of injecting serotonin (Author response image 1) and that the physiological changes in PNs (bursting spontaneous activity, maintenance of temporal firing patterns, and increase odor-evoked responses) persists when the cells are held for longer duration (i.e. 3 hours akin to our behavioral experiments). It is worth noting that 3-hour in-vivo intracellular recordings are not easily achievable and come with many experimental constraints. So far, we have managed to record from two PNs that were held for this long and add them to this rebuttal to support our conclusions. (Author response image 2).

**Author response image 1. sa3fig1:** Palp-opening response (POR) patterns to different odorants remain consistent following serotonin introduction. The probability of PORs is shown as a bar plot for four different odorants; hexanol (green), benzaldehyde (blue), linalool (red), and ammonium (purple). PORs before serotonin injection (solid bars) are compared against response levels after serotonin injection (striped bars). As can be noted, PORs to the four odorants remain consistent when tested 15 minutes and 3 hours after (5HT) serotonin injection.

**Author response image 2. sa3fig2:** Spontaneous and odor-evoked responses in individual PNs remain consistent for three hours after serotonin introduction into the recording chamber/bath. (**A**) Representative intracellular recording showing membrane potential fluctuations in a projection neuron (PN) in the antennal lobe. Spontaneous and odor-evoked responses to four odorants (pink color bars, 4 s duration) are shown before (control) and after serotonin application (5HT). Voltage traces 30 minutes (30min), 1 hour (1h), 2 hours (2h), and 3 hours (3h) after 5HT application are shown to illustrate the persisting effect of serotonin during spontaneous and odor-evoked activity periods. (**B**) Rasterized spiking activities in two recorded PNs are shown. Spontaneous and odor-evoked responses are shown in all 5 consecutive trials. Note that the odor-evoked response patterns are maintained, but the spontaneous activity patterns are altered after serotonin introduction.

We thank the reviewer for again providing very useful feedback for improving our manuscript.

**Reviewer #2 (Public Review):**
Summary:The authors investigate the influence of serotonin on feeding behavior and electrophysiological responses in the antennal lobe of locusts. They find that serotonin injection changes behavior in an odor-specific way. In physiology experiments, they can show that projection neurons in the antennal lobe generally increase their baseline firing and odor responses upon serotonin injection. Using a modeling approach the authors propose a framework on how a general increase in antennal lobe output can lead to odor-specific changes in behavior.Strengths:This study shows that serotonin affects feeding behavior and odor processing in the antennal lobe of locusts, as serotonin injection increases activity levels of projection neurons. This study provides another piece of evidence that serotonin is a general neuromodulator within the early olfactory processing system across insects and even phyla.Weaknesses:I still have several concerns regarding the generalizability of the model and interpretation of results. The authors cannot provide evidence that serotonin modulation of projection neurons impacts behavior.

This is true and likely to be true for any study linking neural responses to behavior. There are multiple circuits and pathways that would get impacted by a neuromodulator like serotonin. What we showed with our physiology is how spontaneous and odor-evoked responses in the very first neural network that receives olfactory sensory neuron input are altered by serotonin. Given the specificity of the changes in behavioral outcomes (i.e. odor-specific increase and decrease in an appetitive behavior) and non-specificity in the changes at the level of individual PNs (general increase in odor-evoked spiking activity), we presented a relatively simple computational model to address the apparent mismatch between neural and behavioral responses (Author response image 4).

The authors show that odor identity is maintained after 5-HT injection, however, the authors do not show if PN responses to different odors were differently affected after serotonin exposure.

The PN responses to different odorants changed in a qualitatively similar fashion. (Author response image 3)

**Author response image 3. sa3fig3:** PN activity before and after 5HT application are compared for different cellodor combinations. As can be noted, the changes are qualitatively similar in all cases. After 5HT application, the baseline activity became more bursty, but the odor-evoked response patterns were robustly maintained for all odorants.

Regarding the model, the authors show that the model works for odors with non-overlapping PN activation. However, only one appetitive, one neutral, and one aversive odor has been tested and modeled here. Can the fixed-weight model also hold for other appetitive and aversive odors that might share more overlap between active PNs? How could the model generate BZA attraction in 5-HT exposed animals (as seen in behavior data in Figure 1) if the same PNs just get activated more?

See Author response image 4.

**Author response image 4. sa3fig4:** Testing the generality of the proposed computational model. To test the generality of the model proposed we used a published dataset (Chandak and Raman, 2023): Neural dataset – 89 PN responses to a panel of twenty-two odorants; Behavioral dataset – probability of POR responses to the same twenty-two odorants. We built the model using just the three odorants overlapping between the two datasets: hexanol, benzaldehyde and linalool. The true probability of POR values of the twenty odorants and the POR probability predicted by the model are shown for all twenty-two odorants as a scatter plot. As can be noted, there is a high correlation (0.79) between the true and the predicted values.

The authors should still not exclude the possibility that serotonin injections could affect behavior via modulation of other cell types than projection neurons. This should still be discussed, serotonin might rather shut down baseline activation of local inhibitory neurons - and thus lead to the interesting bursting phenotypes, which can also be seen in the baseline response, due to local PN-to-LN feedback.

As we agreed, there could be other cells that are impacted by serotonin release. Our goal in this study was to characterize how spontaneous and odor-evoked responses in the very first neural network that receives olfactory sensory neuron input are altered by serotonin. Within this circuit, there are local inhibitory neurons (LNs), as correctly indicated by this reviewer. Surprisingly, our preliminary data indicates that LNs are not shut down but also have an enhanced odor-evoked neural response (Author response image 5). Further data would be needed to verify this observation and determine the mechanism that mediate the changes in PN excitability. Irrespective, since PN activity should incorporate the effects of changes in the local neuron responses and is the sole output from the antennal lobe that drives all downstream odor-evoked activity, we focused on them in this study.

**Author response image 5. sa3fig5:** Representative traces showing intracellular recording from a local neuron in the antennal lobe. Five consecutive trials are shown. Note that LNs in the locust antennal lobe are non-spiking. The LN activity before, during, and after the presentation of benzaldehyde and hexanol (colored bar; 4s) are shown. The Left and Right panels show LN activity before and after the application of 5HT. As can be noted, 5HT did not shut down odor-evoked activity in this local neuron.

The authors did not fully tone down their claims regarding causality between serotonin and starved state behavioral responses. There is no proof that serotonin injection mimics starved behavioral responses.Specific minor issues:It is still unclear how naturalistic the chosen odor concentrations are. This is especially important as behavioral responses to different concentrations of odors are differently modulated after serotonin injection (Figure 2: Linalool and Ammonium). The new method part does not indicate the concentrations of odors used for electrophysiology.

All odorants were diluted to 0.01-10% concentration by volume in either mineral oil or distilled water. This information is included in the Methods section. For most odorants used in the study, the lower concentrations only evoked a very weak neural response, and the higher concentrations evoked more robust responses. The POR responses for these odorants at various concentrations chosen are included in Figure 2. Note, that the responses to linalool and ammonium remained weak throughout the concentration changes, compared to hexanol and benzaldehyde.

Did all tested PNs respond to all odorants?

No, only a subset of them responses to each odorant. These responses have been well characterized in earlier publications [included refs].

The authors do not show if PN responses to different odors were differently affected after serotonin exposure. They describe that ON responses were robust, but OFF responses were less consistent after 5-HT injection. Was this true across all odors tested? Example traces are shown, but the odor is not indicated in Figure 4A. Figure 4D shows that many odor-PN combinations did not change their peak spiking activity - was this true across odorants? In Figure 5 - are PNs ordered by odor-type exposure?Also, Figure 6A only shows example trajectories for odorants - how does the average look? Regarding the data used for the model - can the new dataset from the 82 odor-PN pairs reproduce the activation pattern of the previously collected dataset of 89 pairs?

What is shown in Figure 6A is the trial-averaged response trajectory combining activities of all 82 odor-PN pairs. 82 odor-PN pair was collected intracellularly examining the responses to four odorants before and after 5HT application. The second dataset involving 89 PN responses to 22 odorants was collected extracellularly. They have qualitative similarities in each odorant activate a unique subset of those neurons.

The authors toned down their claims that serotonin injection can mimic the starved state behavioral response. However, some sentences still indicate this finding and should also be toned down:last sentence of introduction - "In sum, our results provide a more systems-level view of how a specific neuromodulator (serotonin) alters neural circuits to produce flexible behavioral outcomes."

We believe we showed this with our computational model, how uniform changes in the neural responses could lead to variable and odor-specific changes in behavioral PORs.

discussion: "Finally, fed locusts injected with serotonin generated similar appetitive responses to food-related odorants as starved locusts indicating the role of serotonin in hunger statedependent modulation of odor-evoked responses." This claim is not supported.

Figure 7 shows that the fed locusts had lower POR to hex and bza. The POR responses significantly increased after the 5HT application. However, we have rephrased this sentence to limit our claims to this result. "Finally, fed locusts injected with serotonin generated similar appetitive palp-opening responses to food-related odorants as observed in starved locusts.”

last results: "However, consistent with results from the hungry locusts, the introduction of serotonin increased the appetitive POR responses to HEX and BZA. Intriguingly, the appetitive responses of fed locusts treated with 5HT were comparable or slightly higher than the responses of hungry locusts to the same set of odorants."

Again this sentence simply describes the result shown in Figure 7.

In Figure 7 - BZA response seems unchanged in hungry and fed animals and only 5-HT injection enhances the response. There is only one example where 5-HT application and starvation induce the same change in behavior - N=1 is not enough to conclude that serotonin influences food-driven behaviors.

The reviewer is ignoring the lack of changes to PORs to linalool and ammonium. Taken together, serotonin increased PORs to only two of the four odorants in starved locusts. The responses after 5HT modulation to these four odorants were similar in fed locusts treated with 5HT and starved locusts.

Also, this seems to be wrongly interpreted in Figure 7: "It is worth noting that responses to LOOL and AMN, non-food related odorants with weaker PORs, remained unchanged in fed locusts treated with 5HT." The authors indicate a significant reduction in POR after 5-HT injection on LOOL response in Figure 7.

Revised.

"It is worth noting that responses to LOOL and AMN, non-food related odorants with weaker PORs, and reduced in fed locusts treated with 5HT."

Also, the newly added sentence at the end of the discussion does not make sense: "However, since 5HT increased behavioral responses in both fed and hungry locusts, the precise role of 5HT modulation and whether it underlies hunger-state dependent modulation of appetitive behavior still remains to be determined."The authors did not test 5-HT injection in starved animals

The results shown in Figure 1 compare the POR responses of starved locusts before and after 5HT introduction.

We again thank the reviewer for useful feedback to further improve our manuscript.

[The following is the authors’ response to the original reviews.]

**Reviewer #1 (Public Review):**
Summary:This manuscript explores the impact of serotonin on olfactory coding in the antennal lobe of locusts and odor-evoked behavior. The authors use serotonin injections paired with an odor-evoked palp-opening response assay and bath application of serotonin with intracellular recordings of odor-evoked responses from projection neurons (PNs).Strengths:The authors make several interesting observations, including that serotonin enhances behavioral responses to appetitive odors in starved and fed animals, induces spontaneous bursting in PNs, and uniformly enhances PN responses to odors. Overall, I had no technical concerns.Weaknesses:While there are several interesting observations, the conclusions that serotonin enhanced sensitivity specifically and that serotonin had feeding-state-specific effects, were not supported by the evidence provided. Furthermore, there were other instances in which much more clarification was needed for me to follow the assumptions being made and inadequate statistical testing was reported.Major concerns.To enhance olfactory sensitivity, the expected results would be that serotonin causes locusts to perceive each odor as being at a relatively higher concentration. The authors recapitulate a classic olfactory behavioral phenomenon where higher odor concentrations evoke weaker responses which is indicative of the odors becoming aversive. If serotonin enhanced the sensitivity to odors, then the dose-response curve should have shifted to the left, resulting in a more pronounced aversion to high odor concentrations. However, the authors show an increase in response magnitude across all odor concentrations. I don't think the authors can claim that serotonin enhances the behavioral sensitivity to odors because the locusts no longer show concentration-dependent aversion. Instead, I think the authors can claim that serotonin induces increased olfactory arousal.

The reviewer makes a valid point. Bath application of serotonin increased POR behavioral responses across all odor concentrations, and concentration-dependent aversion was also not observed. Furthermore, the monotonic relationship between projection neuron responses and the intensity of current injection is altered when serotonin is exogenously introduced (see Author response image 1; see below for more explanation). Hence, our data suggests that serotonin alters the dose-response relationship between neural/behavioral responses and odor intensity. As recommended, we have followed what the reviewer has suggested and revised our claim to serotonin inducing increase in olfactory arousal. The new physiology data has been added as Supplementary Figure 3 to the revised manuscript.

The authors report that 5-HT causes PNs to change from tonic to bursting and conclude that this stems from a change in excitability. However, excitability tests (such as I/V plots) were not included, so it's difficult to disambiguate excitability changes from changes in synaptic input from other network components.

To confirm that the PN excitability did indeed change after serotonin application, we performed a new set of current-clamp recordings. In these experiments, we monitored the spiking activities in individual PNs as we injected different levels of current injections (200 – 1000 pico Amperes). Note that locust LNs that provide recurrent inhibition arborize and integrate inputs from a large number of sensory neurons and projection neurons. Therefore, activating a single PN should not activate the local neurons and therefore the antennal lobe network.

We found that the total spiking activity monotonically increased with the magnitude of the current injection in all four PNs recorded (Author response image 6). However, after serotonin injection, we found that the spiking activity remained relatively stable and did not systematically vary with the magnitude of the current injection. While the changes in odor-evoked responses may incorporate both excitability changes in individual PNs and recurrent feedback inhibition through GABAergic LNs, these results from our current injection experiments unambiguously indicate that there are changes in excitability at the level of individual PNs. We have added this result to the revised manuscript.

**Author response image 6. sa3fig6:** Current-injection induced spiking activity in individual PNs is altered after serotonin application. (A) Representative intracellular recordings showing membrane potential fluctuations as a function of time for one projection neuron (PNs) in the locust antennal lobe. A two-second window when a positive 200-1000pA current was applied is shown. Firing patterns before (left) and after (right) serotonin application are shown for comparison. Note, the spiking activity changes after the 5HT application. The black bar represents the 20mV scale. (B) Dose-response curves showing the average number of action potentials (across 5 trials) during the 2second current pulse before (green) and after (purple) serotonin for each recorded PN. Note that the current intensity was systematically increased from 200 pA to 1000 pA. The (C) The mean number of spikes across the four recorded cells during current injection is shown. The color progression represents the intensity of applied current ranging 200pA (leftmost bar) to 1000pA (rightmost bar). The dose-response trends before (green) and after (purple) 5HT application are shown for comparison.. The error bars represent SEM across the four cells.

There is another explanation for the theoretical discrepancy between physiology and behavior, which is that odor coding is further processing in higher brain regions (ie. Other than the antennal lobe) not studied in the physiological component of this study. This should at least be discussed.

This is a valid argument. For our model of neural mapping onto behavior to work, we only need the odorant that evokes or suppresses PORs to activate a distinct set of neurons. Having said that, our extracellular recording results (Fig. 6E) indicate that hexanol (high POR) and linalool (low POR) do activate highly non-overlapping sets of PNs in the antennal lobe. Hence, our results suggest that the segregation of neural activity based on behavioral relevance already begins in the antennal lobe. We have added this clarification to the discussion section.

The authors cannot claim that serotonin underlies a hunger state-dependent modulation, only that serotonin impacts responses to appetitive odors. Serotonin enhanced PORs for starved and fed locusts, so the conclusion would be that serotonin enhances responses regardless of the hunger state. If the authors had antagonized 5-HT receptors and shown that feeding no longer impacts POR, then they could make the claim that serotonin underlies this effect. As it stands, these appear to be two independent phenomena.

This is also a valid point. We have clarified this in the revised manuscript.

**Reviewer #2 (Public Review):**
Summary:The authors investigate the influence of serotonin on feeding behavior and electrophysiological responses in the antennal lobe of locusts. They find that serotonin injection changes behavior in an odorspecific way. In physiology experiments, they can show that antennal lobe neurons generally increase their baseline firing and odor responses upon serotonin injection. Using a modeling approach the authors propose a framework on how a general increase in antennal lobe output can lead to odorspecific changes in behavior. The authors finally suggest that serotonin injection can mimic a change in a hunger state.Strengths:This study shows that serotonin affects feeding behavior and odor processing in the antennal lobe of locusts, as serotonin injection increases activity levels of antennal lobe neurons. This study provides another piece of evidence that serotonin is a general neuromodulator within the early olfactory processing system across insects and even phyla.Weaknesses:I have several concerns regarding missing control experiments, unclear data analysis, and interpretation of results.A detailed description of the behavioral experiments is lacking. Did the authors also provide a mineral oil control and did they analyze the baseline POR response? Is there an increase in baseline response after serotonin exposure already at the behavioral output level? It is generally unclear how naturalistic the chosen odor concentrations are. This is especially important as behavioral responses to different concentrations of odors are differently modulated after serotonin injection (Figure 2: Linalool and Ammonium).

POR protocol: Sixth instar locusts (Schistocera americana) of either sex were starved for 24-48 hours before the experiment or taken straight from the colony and fed blades of grass for the satiated condition. Locusts were immobilized by placing them in the plastic tube and securing their body with black electric tape (see Author response image 7). Locusts were given 20 - 30 minutes to acclimatize after placement in the immobilization tube. As can be noted, the head of the locusts along with the antenna and maxillary palps protruded out of this immobilization tube so they can be freely moved by the locusts. Note that the maxillary palps are sensory organs close to the mouth parts that are used to grab food and help with the feeding process.

**Author response image 7. sa3fig7:** Pictures showing the behavior experiment setup and representative palp-opening responses in a locust.

It is worth noting that our earlier studies had shown that the presentation of ‘appetitive odorants’ triggers the locust to open their maxillary palps even when no food is presented (Saha et al., 2017; Nizampatnam et al., 2018; Nizampatnam et al., 2022; Chandak and Raman, 2023.) Furthermore, our earlies results indicate that the probability of palp opening varies across different odorants (Chandak and Raman, 2023). We chose four odorants that had a diverse range of palp-opening: supra-median (hexanol), median (benzaldehyde), and sub-median (linaool). Therefore, each locust in our experiments was presented with one concentration of four odorants (hexanol, benzaldehyde, linalool, and ammonium) in a pseudorandomized order. The odorants were chosen based on our physiology results such that they evoked different levels of spiking activities.

The odor pulse was 4 s in duration and the inter-pulse interval was set to 60 s. The experiments were recorded using a web camera (Microsoft) placed right in front of the locusts. The camera was fully automated with the custom MATLAB script to start recording 2 seconds before the odor pulse and end recording at odor termination. An LED was used to track the stimulus onset/offset. The POR responses were manually scored offline. Responses to each odorant were scored a 0 or 1 depending on if the palps remained closed or opened. A positive POR was defined as a movement of the maxillary palps during the odor presentation time window as shown on the locust schematic (Main Paper Figure 1).

As the reviewer inquired, we performed a new series of POR experiments, where we explored POR responses to mineral oil and hexanol, before and after serotonin injection. For this study, we used 10 locusts that were starved 24-48 hours before the experiment. Note that hexanol was diluted at 1% (v/v) concentration in mineral oil. Our results reveal that locusts PORs to hexanol (~ 50% PORs) were significantly higher than those triggered by mineral oil (~10% PORs). Injection of serotonin increased the POR response rate to hexanol but did not alter the PORs evoked by mineral oil (Author response image 8).

**Author response image 8. sa3fig8:** Serotonin does not alter the palp-opening responses evoked by paraffin oil. The PORs before and after (5HT) serotonin injection are summarized and shown as a bar plot for hexanol and paraffin oil. Striped bars signify the data collected after 5HT injection. Significant differences are identified in the plot one-tailed paired-sample t-test; (*p<0.05).

Regarding recordings of potential PNs - the authors do not provide evidence that they did record from projection neurons and not other types of antennal lobe neurons. Thus, these claims should be phrased more carefully.

In the locust antennal lobe, only the cholinergic projection neurons fire full-blown sodium spikes. The GABAergic local neurons only fire calcium ‘spikelets’ (Laurent, TINS, 1996; Stopfer et al., 2003; see Author response image 9 for an example). Hence, we are pretty confident that we are only recording from PNs. Furthermore, due to the physiological properties of the LNs, their signals being too small, they are also not detected in the extracellular recordings from the locust antennal lobe. Hence, we are confident with our claims and conclusion.

**Author response image 9. sa3fig9:** PN vs LN physiological differences: Left: A representative raw voltage traces recorded from a local neuron before, during, and after a 4-second odor pulse are shown. Note that the local neurons in the locust antennal lobe do not fire full-blown sodium spikes but only fire small calcium spikelets. On the right: A representative raw voltage trace recorded from a representative projection neuron is shown for comparison. Clear sodium spikes are clearly visible during spontaneous and odor-evoked periods. The gray bar represents 4 seconds of odor pulse. The vertical black bar represents the 40mV.

The presented model suggests labeled lines in the antennal lobe output of locusts. Could the presented model also explain a shift in behavior from aversion to attraction - such as seen in locusts when they switch from a solitarious to a gregarious state? The authors might want to discuss other possible scenarios, such as that odor evaluation and decision-making take place in higher brain regions, or that other neuromodulators might affect behavioral output. Serotonin injections could affect behavior via modulation of other cell types than antennal lobe neurons. This should also be discussed - the same is true for potential PNs - serotonin might not directly affect this cell type, but might rather shut down local inhibitory neurons.

There are multiple questions here. First, regarding solitary vs. gregarious states, we are currently repeating these experiments on solitary locusts. Our preliminary results (not included in the manuscript) indicate that the solitary animals have increased olfactory arousal and respond with a higher POR but are less selective and respond similarly to multiple odorants. We are examining the physiology to determine whether the model for mapping neural responses onto behavior could also explain observations in solitary animals.

Second, this reviewer makes the point raised by Reviewer 1. We agree that odor evaluation and decisionmaking might take place in higher brain regions. All we could conclude based on our data is that a segregation of neural activity based on behavioral relevance might provide the simplest approach to map non-specific increase in stimulus-evoked neural responses onto odor-specific changes in behavioral outcome. Furthermore, our results indicate that hexanol and linalool, two odorants that had an increase and decrease in PORs after serotonin injection, had only minimal neural response overlap in the antennal lobe. These results suggest that the formatting of neural activity to support varying behavioral outcomes might already begin in the antennal lobe. We have added this to our discussion.

Third, regarding serotonin impacting PNs, we performed a new set of current-clamp experiments to examine this issue (Author response image 1). Our results clearly show that projection neuron activity in response to current injections (that should not incorporate feedback inhibition through local neurons) was altered after serotonin injection. Therefore, the observed changes in the odor-evoked neural ensemble activity should incorporate modulation at both individual PN level and at the network level. We have added this to our discussion as well.

Finally, the authors claim that serotonin injection can mimic the starved state behavioral response. However, this is only shown for one of the four odors that are tested for behavior (HEX), thus the data does not support this claim.

We note that Hex is the only appetitive odorant in the panel. But, as reviewer 1 has also brought up a similar point, we have toned down our claims and will investigate this carefully in a future study.

**Recommendations for the authors:**

**Reviewer #1 (Recommendations For The Authors):**
Was the POR of the locusts towards linalool and ammonium higher than towards a blank odor cartridge? I ask because the locusts appear to be less likely to respond to these odors and so I am concerned that this assay is not relevant to the ecological context of these odors. In other words, perhaps serotonin did not enhance the responses to these odors in this assay, because this is not a context in which locusts would normally respond to these odors.

The POR response to linalool and ammonium is lower and comparable to that of paraffin oil. Serotonin does not increase POR responses to paraffin oil but does increase response to hexanol (an appetitive odorant). We have clarified this using new data (Author response image 10).

**Author response image 10. sa3fig10:** Odor-evoked responses of four PNs that received positive weights in the model (top panel), and four PNs that were assigned negative weights in the model (bottom).

It seems to me that Figure 5C is the crux for understanding the potential impact of 5-HT on odor coding, but it is somewhat confusing and underutilized. Is the implication that 5-HT decorrelates spontaneous activity such that when an odor stimulus arrives, the odor-evoked activity deviates to a greater degree? The authors make claims about this figure that require the reader to guess as to the aspect of the figure to which they are referring.

The reviewer makes an astute observation. Yes, the spontaneous activity in the antennal lobe network before serotonin introduction is not correlated with the ensemble spontaneous activity after serotonin bath application. Remarkably, the odor-evoked responses were highly similar, both in the reduced PCA space and when assayed using high-dimensional ensemble neural activity vectors. Whether the changes in network spontaneous activity have a function in odor detection and recognition is not fully understood and cannot be convincingly answered using our data. But this is something that we had pondered.

The modeling component summarized in Figure 6 needs clarification and more detail. Perhaps example traces associated with positive weighting within neural ensemble 1 relative to neural ensemble 2? I struggled to understand conceptually how the model resolved the theoretical discrepancy between physiology and behavior.

As recommended, here is a plot showing the responses of four PNs that had positive weights to hexanol and linalool. As can be expected, each PN in this group had higher responses to hexanol and no response to linalool. Further, the four PNs that received negative weights had response only to linalool.

Was there a significant difference between the PORs of hungry vs. fed locusts? The authors state that they differ and provide statistics for the comparisons to locusts injected with 5-HT, but then don't provide any statistical analyses of hungry vs. fed animals.

The POR responses to HEX (an appetitive odorant) were significantly different between the hungry and starved locusts (Author response image 11).

**Author response image 11. sa3fig11:** A bar plot summarizing PORs to all four odors for satiated locust (highlighted with stripes), before (dark shade), and after 5HT injection (lighter shade). To allow comparison before 5HT injection for starved locust plotted as well (without stripes). The significance was determined using a one-tailed paired-sample ttest(*p<0.05).

Were any of the effects of 5-HT on odor-evoked PN responses significant? No statistics are provided.

We examined the distribution of odor-evoked responses in PNs before and after 5HT introduction. We found that the overall distribution was not significantly different between the two (one-tailed pairedsample t-test; p = 0.93) (Author response image 12) .

**Author response image 12. sa3fig12:** Comparison of the distribution of odor-evoked PN responses before (green) and after (purple) 5HT introduction. One-tailed paired sample t-test was used to compare the two distributions.

The authors interchangeably use "serotonin", "5HT" and "5-HT" throughout the manuscript, but this should be consistent.

This has been fixed in the revised manuscript.

On page 2 the authors provide an ecological relevance for linalool as being an additive in pesticides, however, linalool is a common floral volatile chemical. Is the implication that locusts have learned to associate linalool with pesticides?

Linalool is a terpenoid alcohol that has a floral odor but has also been used as a pesticide and insect repellent [Beier et al., 2014]. As shown in Author response image 2, it evoked the least POR responses amongst a diverse panel of 22 odorants that were tested. We have clarified how we chose odorants based on the prior dataset in the Methods section.

In Figure 1, there should be a legend in the figure itself indicating that the black box indicates the absence of POR and the white box indicates presence, rather than just having it in the legend text.

Done.

In Figure 2, the raw data from each animal can be moved to the supplements. The way it is presented is overwhelming and the order of comparisons is difficult to follow.

Done.

For the induction of bursting in PNs by the application of 5-HT, were there any other metrics observed such as period, duration of bursts, or peak burst frequency? The authors rely on ISI, but there are other bursting metrics that could also be included to understand the nature of this observation. In particular, whether the bursts are likely due to changes in intrinsic biophysical properties of the PNs or polysynaptic effects.

We could use other metrics as the reviewer suggests. Our main point is that the spontaneous activity of individual PNs changed. We have added a new current-injection experiments to show that the PNs output to square pulses of current becomes different after serotonin application (Author response image 1)

Were 4-vinyl anisole, 1-nonanol, and octanoic acid selected as additional odors because they had particular ecological relevance, or was it for the diversity of chemical structure?

These odorants were selected based on both, chemical structure and ecological relevance. The logic behind this was to have a very diverse odor panel that consisted of food odorant – Hexanol, aggregation pheromone – 4-vinyl anisole, sex pheromone – benzaldehyde, acid – octanoic acid, base – ammonium, and alcohol – 1-nonanol. Additionally, we selected these odors based on previous neural and behavioral data on these odorants (Chandak and Raman, 2023, Traner and Raman, 2023, Nizampatnam et al, 2022 & 2018; Saha et al., 2017 & 2013).

**Reviewer #2 (Recommendations For The Authors):**
The electrophysiology dataset combines all performed experiments across all tested different PN-odor pairs. How many odors have been tested in a single PN and how many PNs have been tested for a single odor? This information is not present in the current manuscript. Can the authors exclude that there are odor-specific modulations?

In total, our dataset includes recordings from 19 PNs. Seven PNs were tested on a panel of seven odorants (4-vinyl anisole, 1-nonanol, octanoic acid, Hex, Bza, Lool, and Amn), and the remaining twelve were tested with the four main odorants used in the study (Hex, Bza, Lool, and Amn). This information has been added to the Methods section

How did the authors choose the concentrations of serotonin injections and bath applications - is this a naturalistic amount?

The serotonin concentration for ephys experiments was chosen based on trial-error experiments:

0.01mM was the highest concentration that did not cause cell death. For the behavioral experiments, we increased the concentration (0.1 M) due to the presence of anatomical structures in the locust's head such as air sacks, sheath as well as hemolymph which causes some degree of dilution that we cannot control.

Behavior experiments were performed 3 hours after injection - ephys experiments 5-10 minutes following bath application. Can the authors exclude that serotonin affects neural processing differently on these different timescales?

We cannot exclude this possibility. We did ePhys experiments 5-10 minutes after bath application as it would be extremely hard to hold cells for that long.

A longer delay was required for our behavioral experiments as the locusts tended to be a bit more agitated with larger spontaneous movements of palps as well as exhibited unprompted vomiting. A 3hour period allowed the locust to regain its baseline level movements after 5HT introduction. [This information has been added to the methods section of the revised manuscript]

Concerning the analysis of electrophysiological data. The authors should correct for changes in the baseline before performing PCA analysis. And how much of the variance is explained by PC1 and PC2?

We did not correct for baseline changes or subtract baseline as we wanted to show that the odor-evoked neural responses still robustly encoded information about the identity of the odorant.

The authors should perform dye injections after recordings to visualize the cell type they recorded from. Serotonin might affect also other cell types in the antennal lobe.

As mentioned above, in the locust antennal lobe only PNs fire full-blown sodium spikes, and LNs only fire calcium spikelets (Author response image 4). Since these signals are small, they will be buried under the noise floor when using extracellular recording electrodes for monitoring responses in the AL antennal lobe.

Hence we are pretty certain what type of cells we are recording from.

There were several typos in the manuscript, please check again.

We have fixed many of the grammatical errors and typos in the revised version.